# Evaluation of rapid diagnostic test kits for detection of *Treponema pallidum* antibody

**Sirinart Chomean**[1,2], **Palakorn Puttaruk**[3], **Phakawat Khamsophar**[3], **Waraphorn Fukpo**[3], **Chollanot Kaset**[1,2] *

1 Department of Medical Technology, Faculty of Allied Health Sciences, Thammasat University, Pathum Thani, Thailand, 2 Thammasat University Research Unit in Medical Technology and Precision Medicine Innovation, Thammasat University, Pathum Thani, Thailand, 3 Medical Technology Laboratory, Thammasat University Hospital, Thammasat University, Pathum Thani, Thailand

* chollanotk@gmail.com

**Data Availability Statement:** ll relevant data are within the manuscript and its Supporting Information files.

## Abstract

Rapid syphilis testing plays a crucial role in global health strategies, addressing the urgent need for prompt and accurate diagnostics, especially in settings with limited resources. Despite their practical utility, these tests often lack thorough validation, leading to concerns about their efficacy and reliability. This study aims to evaluate two prototypes of the Onsite Syphilis Ab Combo Rapid Test (Fd and Ff) and compare their performance with the established chemiluminescent microparticle immunoassay (CMIA) method. Employing a reverse algorithm approach, the study analyzed 450 serum samples, including those from syphilis patients, healthy individuals, and cases with potential cross-reactions. Results of the rapid test kit were then correlated with CMIA findings, RPR, and TPPA titers. The results showed that prototype Fd exhibited a sensitivity of 100.0%, specificity of 98.8%, positive predictive value (PPV) of 8.4%, negative predictive value (NPV) of 100.00% and accuracy of 98.8%. Similarly, prototype Ff exhibited sensitivity of 100.0%, but with a slightly higher specificity of 99.6%, PPV of 21.5%, NPV of 100.0% and accuracy of 99.6%. Moreover, both prototypes Fd and Ff of the Onsite Syphilis Ab Combo Rapid Test demonstrated significant efficacy diagnostic tool, offering clear and straightforward interpretation for clinicians in varied CMIA, RPR and TPPA titer scenarios. The Onsite Syphilis Ab Combo Rapid Test prototypes, Fd and Ff, demonstrated high sensitivity and specificity, comparable to CMIA methods. The effectiveness highlights their suitability for syphilis screening, particularly in non-laboratory settings or situations requiring immediate results. The validation of these prototypes supports their integration into current syphilis diagnostic algorithms, potentially contributing to improved public health outcomes.

## Introduction

Syphilis, a well-known sexually transmitted infection, occurs due to the bacterium *Treponema pallidum* (TP) [1]. Despite advancements in detection methods and affordable, effective treatments, syphilis continues to be a significant health issue. Rapid identification and treatment of

**Funding:** Our research received specific funding from the Thammasat University Research Unit in Medical Technology and Precision Medicine Innovation. Assoc. Prof. Dr. Chollanot Kaset is the principal investigator who received this grant. This funding was also received by Dr. Sirinart Chomean. The funder did not directly influence the study design, data collection and analysis, decision to publish, or preparation of the manuscript.

**Competing interests:** The authors have declared that no competing interest exist.

this infection in its initial stages are crucial to halt its spread and prevent new cases [2, 3]. In the past, syphilis screening predominantly followed a forward or traditional algorithm, starting with non-treponemal tests such as the rapid plasma reagin (RPR) or the Venereal Disease Research Laboratory (VDRL) tests [3–6]. These tests, while efficient for mass screenings, have limitations including the possibility of false positives due to various factors like pregnancy or other inflammatory diseases. Positive results require further confirmation through more specific treponemal tests, such as the *Treponema pallidum* particle agglutination (TPPA) or the fluorescent treponemal antibody absorbed (FTA-ABS) tests [5].

Modern practices are increasingly adopting reverse algorithms. This newer approach initiates the screening process with more sensitive treponemal tests, like enzyme immunoassays (EIA) or chemiluminescent microparticle immunoassays (CMIA), known for reducing false negatives. If these tests are reactive, non-treponemal tests are then used for confirmation. One advantage of the reverse algorithm is the enhanced detection of latent or treated syphilis cases that might be missed by non-treponemal tests [3, 6]. However, a potential downside is the increased cost and the challenge of interpreting results from individuals who might have received treatment in the past [7].

For high-throughput laboratories, there is an emerging predilection for the reverse algorithm, catalyzed by the automation of testing procedures. The BioPlex 2200 Syphilis system, predicated on Multiplex Flow technology, has garnered attention for its potential efficacy. Comparative analyses between BioPlex 2200 Syphilis IgG and IgM assays and the Architect Syphilis TP have illuminated the superior specificity of the former, notwithstanding equivalent sensitivity metrics [7]. These insights suggest that the BioPlex 2200 system, with its automated capabilities and high processing volume, could emerge as a preferential option for syphilis testing in large-scale and high-throughput scenarios. The adjunctive utilization of the IgM assay augments the diagnostic acumen for active syphilitic infections. Recent evaluations of rapid tests underscore their transformative potential in syphilis diagnosis and control, particularly in resource-constrained settings [8, 9]. These studies have illuminated both the strengths and areas for improvement in current rapid test, providing a pathway for enhanced public health intervention.

Herring et al. (2006) conducted a comprehensive evaluation of nine rapid tests, revealing that all tests demonstrated commendable performance relative to traditional methods, with sensitivities and specificities ranging from 84.5–97.7% and 84.5–98%, respectively [8]. However, they noted variability in result stability, particularly when readings were delayed beyond the recommended timeframe, emphasizing the need for strict adherence to test protocols [8]. In a similar study, Diaz et al. (2004) focused on the Determine™ Syphilis TP kit, highlighting its high sensitivity (95.6 to 98.4%) and specificity (95.7 to 97.3%), with minimal inter-reader variability [9]. The assay's performance remained robust across different subsets of patients, including those with HIV. These findings suggest the assay's suitability for diverse patient populations, even outside conventional laboratory settings. Bazzo et al. (2017) evaluated seven commercially available rapid tests in Brazil, with all but one meeting the country's Ministry of Health's minimum specificity requirements [10]. These tests showed sensitivity ranging from 94.5% to 100% and specificity between 91.5% and 100%, affirming their reliability. The study also highlighted the tests' operational efficiency, a critical factor in real-world health service delivery [10]. However, not all rapid tests have performed consistently well across different studies. Herring et al. (2006) found that the Determine™ Syphilis TP kit, when used with whole blood, showed significantly lower sensitivity compared to laboratory-based studies using serum. This finding indicates that while rapid tests are generally effective, their performance can be influenced by the type of specimen used and the testing environment [8].

The evolution in syphilis testing is further marked by the introduction of rapid syphilis tests, which are particularly beneficial in environments lacking sophisticated diagnostic

equipment [4–6]. These rapid tests are prized for their user-friendliness, storage convenience, and direct results interpretation, though they faced regulatory scrutiny due to quality control concerns.

Recognizing these challenges, our study aims to assess the accuracy and reliability of two new versions (Fd and Ff) of The Onsite Syphilis Ab Combo Rapid Test. These innovative tests use lateral flow immunoassay technology and were rigorously evaluated against the benchmark of the Abbott Architect i2000SR Immunoassay Analyzer, a standard CMIA protocol for the qualitative discernment of antibodies (IgG and IgM) against TP. This inquiry sought to ascertain the fidelity and consistency of these innovative rapid tests, assessing their potential as integral components in the multifaceted global strategy against syphilis.

## Materials and methods

### Study population

This research received ethical approval from the Human Research Ethics Committee of Thammasat University (HREC-TUSc) under the approval number 66AH025 (Supporting information: Ethics approval document). The collection of serum samples and CMIA data commenced following the receipt of ethical approval, with the collection period spanning from July 4, 2023, to August 30, 2023. A total of 450 serum samples were collected from Thammasat University Hospital (TUH). These were classified into three groups: 200 reactive samples for T. pallidum antibodies obtained from syphilis patients, 200 non-reactive samples from healthy individuals, and 50 non-reactive samples from pregnant women and individuals with conditions such as HIV, HBV, and HCV, which have the potential to cause cross-reactions. Specifically, 200 samples were obtained from the positive group using the CMIA method, following standard clinical diagnostic procedures that included clinical assessment and serological tests such as the rapid plasma reagin (RPR) test and Treponema pallidum particle agglutination assay (TPPA). All serum samples were processed and analyzed using the reverse algorithm procedure.

### Methods

**The Onsite Syphilis Ab Combo Rapid Test.** This study was designed to assess the effectiveness of two rapid screening kits, the Onsite Syphilis Ab Combo Rapid Test, prototypes Fd and Ff, utilizing the lateral flow chromatographic immunoassay method for detection. These kits aim to identify IgM, IgG, and IgA antibodies against the recombinant antigen of the Treponema pallidum (Tp) bacterium. For each test, 40 μL of serum was mixed with 40 μL of the provided buffer. After a 15-minute incubation at room temperature, the results were visually interpreted. Samples that showed a red line at the control band but no line at the test band were considered non-reactive. In contrast, samples that displayed red lines at both the test and control bands were classified as reactive. Reactive samples underwent further evaluation using a non-Treponemal method.

The test cassette operates on the principle of detecting antibodies through a colored conjugate pad containing recombinant Tp antigens conjugated with colloidal gold (Tp conjugates), alongside a control antibody also linked to colloidal gold. This setup comprises a nitrocellulose membrane strip with two distinct lines: a test line (T line) and a control line (C line). The T line is precoated with non-conjugated recombinant Tp antigens, serving to capture the specific syphilis antibodies from the sample if present, while the C line is precoated with a control line antibody, ensuring the test is performed correctly. When a sample containing Tp antibodies is applied, it reacts with the Tp antigen-gold conjugates. This mixture migrates along the strip by capillary action, interacting with the antigen coated on the T line, forming a colored line,

indicating a reactive result. The appearance of the colored line at the C line serves as a procedural control, confirming sufficient sample volume and correct procedural technique.

**Chemiluminescent microparticle immunoassay (CMIA).** The efficiency of these two Onsite Syphilis Ab Combo Rapid Test prototypes was evaluated against the benchmark of the Abbott Architect i2000SR Immunoassay Analyzer (Abbott, Illinois, U.S.A.), a standard CMIA protocol for the qualitative discernment of antibodies (IgG and IgM) against TP. Architect i2000SR operates as a two-phase immunoassay, the reaction principle is as follows: Microparticle is coated with three recombinant antigens (Ag) (TpN15, TpN17, TpN47) to bind IgG and IgM antibodies specific for TP in human serum or plasma. Results from the test were given in a signal to cut-off (S/CO) ratio, with a value greater than 1.0 indicating reactivity sample. Per the manufacturer's documentation, the test's sensitivity stands at 100%, and its specificity at 99.76%.

**Rapid Plasma Reagin (RPR).** Reactive serum samples were analyzed using the RPR method, specifically with RPR-Carbon from Cypress Diagnostics (Hulshout, Belgium). This test employed 50 μL of serum and 20 μL of RPR carbon reagent. Positive and negative controls were also included. Samples were agitated using an orbital shaker for 8 minutes, and results were visually interpreted. Non-reactive samples were advanced to a secondary Treponemal test for confirmation.

**Treponema pallidum particle agglutination (TPPA).** Non-reactive samples from the RPR method were further examined using the conventional Treponemal TPPA test, with the SERODIA®-TPPA kit (Fujirebio Holdings, Inc, Tokyo, Japan). The test incorporated a 1:40 diluted serum, using 25 μL of serum and 25 μL of sensitized particles. After a 2-hour incubation, results were visually interpreted, with reactive samples undergoing further titer assessment.

## Diagnostic performance of Onsite Syphilis Ab Combo Rapid Test

The 450 serum samples were analyzed using the Onsite Syphilis Ab Combo Rapid Test (both Fd and Ff prototypes). The efficiency of these two Onsite Syphilis Ab Combo Rapid Test prototypes was evaluated against a standard CMIA protocol for the qualitative discernment of antibodies (IgG and IgM) against TP, comparing the results against serum samples from syphilis patients, healthy individuals, pregnant women, and individuals with diseases that could potentially cause cross-reactions.

For samples with inconsistent results, e.g., non-reactive in CMIA but reactive in the rapid test or vice versa, further testing was conducted using the Determine™ Syphilis TP kit from Abbott (Abbott, Illinois, U.S.A.). This involved adding 50 μL of serum to the test chamber and allowing it to incubate at room temperature for 15 minutes. Results were visually interpreted.

## Correlation analysis of the Onsite Syphilis Ab Combo Rapid Test with CMIA, RPR and TPPA test

The correlation analysis between the strength of reaction from the Onsite Syphilis Ab Combo Rapid Test and signal/ratio results from the CMIA, RPR titer and TPPA titer were evaluated. We categorized the signal ratio [sample/cut-off (s/co)] of the CMIA for the 200 positive samples into different intervals, five specific ranges were identified: 1–9, 10–19, 20–29, 30–39, and $\geq 40$ reflecting the actual findings in daily screenings. We also categorized the RPR titer into eleven levels: undiluted, 1:2 to 1:1024. Likewise, TPPA titer was categorized into nine levels: 1:40 to 1:10240. Further, the reaction levels from the Onsite Syphilis Ab Combo Rapid Test were grouped into W+, 2+, and 4+ then analyzed with categorized results from CMIA, RPR and TPPA test.

## Statistical data analysis

Statistical analysis was performed using MedCal software (version 18.2.1; MedCalc, Maria-kerke, Belgium). The performance of the Onsite Syphilis Ab Combo Rapid Test included sensitivity, specificity, positive predictive value (PPV), negative predictive value (NPV), positive likelihood ratio (+LR), negative likelihood ratio (-LR) and accuracy was analyzed applying standard formulas [11].

## Results

### Diagnostic performance of Onsite Syphilis Ab Combo Rapid Test

In the evaluation of the Onsite Syphilis Ab Combo Rapid Test (prototype Fd) against the CMIA method, 200 samples (100%) accurately matched the reactive results from CMIA, while 3 samples (1.2%) did not align with the non-reactive CMIA results. On the other hand, of the samples that tested negative with the rapid test, all 247 samples (98.8%) corresponded with the non-reactive outcomes from CMIA. In the assessment of the Onsite Syphilis Ab combo Rapid Test (prototype Ff) in comparison to the CMIA method, of these positive results, 200 samples (100%) were in agreement with the reactive results from the CMIA. However, 1 sample (0.4%) was discordant with the non-reactive outcomes of CMIA. Conversely, among the samples deemed negative by the rapid test, all 249 samples (99.6%) were consistent with the non-reactive findings from CMIA.

The Onsite Syphilis Ab Combo Rapid Test, two prototypes, Fd and Ff, were analyzed for their diagnostic performance across a range of metrics. The performance of the Onsite Syphilis Ab Combo Rapid Test, prototypes Fd and Ff, included sensitivity, specificity, PPV, NPV, +LR, -LR, and accuracy, as illustrated in Table 1. Both prototypes demonstrated exemplary sensitivity, each registering at 100% (95%CI: 98.2–100.0). Specificity, another critical parameter, varied slightly between the two, with prototype Fd exhibiting a specificity of 98.8% (95% CI: 96.5–99.8) which lower than prototype Ff showing at 99.6% (95% CI: 97.8–100.0). The data indicated that both prototypes exhibit high sensitivity and NPV while prototype Ff has a marginal advantage in specificity, PPV, PLR, and overall accuracy (Table 1).

In our study, we observed notable discrepancies in the detection of syphilis antibodies among three distinct samples when comparing two prototypes of The Onsite Syphilis Ab Combo Rapid Test (prototypes Fd and Ff) with results from CMIA, RPR, TPPA, and the Determine™ Syphilis TP kit (S1 Fig). Specifically, sample no. 260, which was non-reactive in CMIA (S/CO ratio = 0.1), RPR, and TPPA tests, paradoxically showed a strongly positive reaction with both prototypes Fd and Ff of the Onsite Syphilis Ab combo Rapid Test, while the Determine™ Syphilis TP kit, RPR and TPPA reported a negative result. Similarly, sample no. 331 displayed non-reactive results in CMIA (S/CO ratio = 0.07), RPR, and TPPA tests and

**Table 1.  Evaluation summary of rapid diagnostic test kits for treponema pallidum antibody detection.**

| Diagnosis Performance | The Onsite Syphilis Ab combo rapid test | |
|---|---|---|
| | Fd | Ff |
| **Sensitivity % (95% Cl)** | 100.00 (98.17–100.00) | 100 (98.17–100.00) |
| **Specificity % (95% Cl)** | 98.80 (96.53–99.75) | 99.60 (97.79–99.99) |
| **PPV % (95% Cl)** | 1.64 (0.54–4.88) | 4.76 (0.70–26.13) |
| **NPV % (95% Cl)** | 100.00 (98.52–100.00) | 100 (98.53–100.00) |
| **PLR (95% Cl)** | 83.33 (27.06–256.63) | 250.00 (35.35–1767.68) |
| **NLR (95% Cl)** | 0 | 0 |
| **Accuracy % (95% Cl)** | 98.00 (97.30–99.59) | 99.60 (98.47–99.96) |

prototype Ff of the Onsite Syphilis Ab combo Rapid Test (prototypes Ff) in contrast to proto-type Fd which indicated a weakly positive reaction, aligning with the negative result of the Determine™ Syphilis TP kit. Lastly, sample no. 332 also non-reactive in CMIA (S/CO ratio = 0.09), RPR, TPPA, and showed a negative result in prototype Ff while a weakly positive in prototype Fd of the Onsite Syphilis Ab combo Rapid Test, with the Determine™ kit reporting negative. These discrepancies, particularly in samples with non-reactive results in CMIA test exceeding a cut-off of 1, highlight the complexities and potential challenges in syphilis anti-body detection, emphasizing the necessity for thorough and multi-faceted diagnostic approaches.

## Correlation analysis of the Onsite Syphilis Ab Combo Rapid Test with CMIA, RPR and TPPA test

To study the proportion and relationship of the intensity of the results from The Onsite Syphi-lis Ab combo Rapid Test in various signal (s/co) ratio of the CMIA method, we organized and divided the s/co ratio of the CMIA method into different ranges, 1.01–10.00, 10.01–20.00, 20.01–30.00, and >30.00. We then categorized the reaction intensity of The Onsite Syphilis Ab combo Rapid Test for each sample into weak positive (+) and strong positive (++).

As showed in Table 2, out of the 200 positive samples assessed, only 1.5% showed a weak positive (+) reaction. A dominating 98.5% exhibited a strong positive (++) reaction, highlight-ing the predominant strong reactivity in the samples assessed using prototype Fd of the Onsite Syphilis Ab combo rapid test. For the prototype Ff, the 200 samples were evaluated which 8.5% of samples exhibited a weak positive (+) reaction, whereas a significant 91.5% demonstrated a strong positive (++) intensity.

In evaluating the Onsite Syphilis Ab combo Rapid Test, our study focused on its perfor-mance across various RPR titer levels, using samples that had positive results in the Onsite Syphilis Ab combo Rapid Test and showed reactive RPR results (n = 176 samples). The analy-sis revealed a consistent pattern of strong positive (++) reactions across the spectrum of RPR titers. Analysis of prototype Fd of the Onsite Syphilis Ab combo Rapid Test demonstrated that at the highest tested titer of >1:1024, both samples analyzed displayed strong positive (++) responses (Table 3). These findings underscore the high efficacy of the Onsite Syphilis Ab combo Rapid test across different RPR titer levels, predominantly yielding strong positive reac-tions, thus highlighting its reliability for syphilis antibody detection in various clinical settings.

In the context of the Onsite Syphilis Ab combo Rapid Test using prototype Ff, Table 3 illus-trated that it performs effectively across various RPR titer levels, predominantly yielding strong

**Table 2. Correlation analysis of reaction strength from the Onsite Syphilis Ab Combo Rapid Test and CMIA results in each s/co interval.**

| CMIA results | Number of sample | The Onsite Syphilis Ab combo rapid test | | | |
|---|---|---|---|---|---|
| | | Fd (%) | | Ff (%) | |
| | | + | ++ | + | ++ |
| 1.01–10.00 | 23 | 1 (4.3%) | 22 (95.7%) | 5 (21.7%) | 18 (78.3%) |
| 10.01–20.00 | 44 | 1 (2.3%) | 43 (97.7%) | 4 (9.1%) | 40 (90.9%)) |
| 20.01–30.00 | 107 | 1 (0.9%) | 106 (99.1%) | 5 (4.7%) | 102 (95.3%) |
| >30.00 | 26 | 0 | 26 (100%) | 3 (11.5%) | 23 (88.5%) |
| Total | 200 (100%) | 3 (1.5%) | 197 (98.5%) | 17 (8.5%) | 183 (91.5%) |

**Table 3. Correlation analysis of reaction strength from the Onsite Syphilis Ab Combo Rapid Test and RPR titer levels.**

| RPR titer | Number of sample | The Onsite Syphilis Ab combo rapid test | | | |
|---|---|---|---|---|---|
| | | Fd (%) | | Ff (%) | |
| | | + | ++ | + | ++ |
| Undiluted | 14 | 0 | 14 (100%) | 1 (7.1%) | 13 (92.9%) |
| 2 | 21 | 0 | 21 (100%) | 0 | 21 (100%) |
| 4 | 24 | 0 | 24 (100%) | 1 (4.2%) | 23 (95.8%) |
| 8 | 32 | 0 | 32 (100%) | 1 (3.1%) | 31 (96.9%) |
| 16 | 21 | 1 (4.8%) | 20 (95.2%) | 2 (9.5%) | 19 (90.5%) |
| 32 | 11 | 1 (9.1%) | 10 (90.9%) | 1 (9.1%) | 10 (90.9%) |
| 64 | 17 | 0 | 17 (100%) | 3 (17.6%) | 14 (82.4%) |
| 128 | 20 | 0 | 20 (100%) | 1 (5.0%) | 19 (95.0%) |
| 256 | 10 | 1 (10.0%) | 9 (90.0%) | 3 (30.0%) | 7 (70.0%) |
| 512 | 4 | 0 | 4 (100%) | 0 | 4 (100%) |
| >1024 | 2 | 0 | 2 (100%) | 1 (50.0%) | 1 (50.0%) |
| Total | 176 (100%) | 3 (1.7%) | 173 (98.3%) | 14 (8.0%) | 162 (92.0%) |

positive reactions, though with occasional weak positive results at certain titer levels. This pattern suggests the prototype's reliability and nuanced sensitivity in detecting syphilis antibodies across a range of clinical scenarios.

We analyzed 24 samples based on their nonreactive RPR results but reactive TPPA results, providing a unique perspective into the performance across a spectrum of TPPA titers. The results revealed that all samples (100%) exhibited strong positive (++) reactions across various TPPA titer levels, indicating the high sensitivity of prototype Fd from the Onsite Syphilis Ab Combo Rapid Test (Table 4). It's noteworthy that no samples were available at the 1:640 TPPA titer, representing a limitation in this titer range within our study. This uniformity in strong positive reactions across diverse TPPA titer levels highlights the efficacy of prototype Fd in detecting syphilis infections, particularly in scenarios where RPR results may not align with TPPA findings. The consistent performance of prototype Fd across varying antibody concentrations, as indicated by the TPPA titers, is indicative of its reliability and potential as a diagnostic tool in syphilis screening.

For prototype Ff, the results indicated a predominantly strong positive (++) reaction (87.5%) across various TPPA titer levels (Table 4). These findings suggest that while prototype Ff is highly sensitive in detecting syphilis infections, particularly in cases with nonreactive RPR and reactive TPPA results, it may show some variability in reaction intensity at specific titer levels. This variability, particularly at the 1:160 and 1:1280 titer levels, could be indicative of the test's nuanced sensitivity to varying antibody concentrations, a factor worth considering in clinical interpretations.

In conclusion, Prototype Fd showed a marginally higher consistency in eliciting strong positive reactions across all TPPA titer levels, suggesting its slightly superior sensitivity in

**Table 4. Correlation analysis of reaction strength from the Onsite Syphilis Ab Combo Rapid Test and TPPA titer levels.**

| TPPA titer | Number of sample | The Onsite Syphilis Ab combo rapid test | | | |
|---|---|---|---|---|---|
| | | Fd (%) | | Ff (%) | |
| | | + | ++ | + | ++ |
| 40 | 1 | 0 | 1 (100%) | 0 | 1 (100%) |
| 80 | 6 | 0 | 6 (100%) | 0 | 6 (100%) |
| 160 | 4 | 0 | 4 (100%) | 1 (25.0%) | 3 (75.0%) |
| 320 | 3 | 0 | 3 (100%) | 0 | 3 (100%) |
| 640 | 0 | 0 | 0 | 0 | 0 |
| 1280 | 7 | 0 | 7 (100%) | 2 (28.6%) | 5 (71.4%) |
| 2560 | 1 | 0 | 1 (100%) | 0 | 1 (100%) |
| 5120 | 1 | 0 | 1 (100%) | 0 | 1 (100%) |
| >10240 | 1 | 0 | 1 (100%) | 0 | 1 (100%) |
| Total | 24 (100%) | 0 | 24 (100%) | 3 (12.5%) | 21 (87.5%) |

detecting syphilis antibodies. Prototype Ff, while also highly effective, presented minor variability at certain titer levels, which could allow for nuanced interpretation in specific clinical scenarios. Both prototypes prove valuable for syphilis screening, with Fd offering a slight advantage in uniform sensitivity, and Ff providing flexibility in response interpretation. This comparative analysis underscores the importance of selecting an appropriate diagnostic tool based on the clinical context and the need for confirmatory testing to ensure diagnostic accuracy.

## Discussion

The diagnosis of syphilis, a significant global health challenge, has witnessed a paradigm shift with the introduction of various serological testing methods each presenting unique benefits and limitations. This evolution in diagnostic approaches is particularly crucial given the diverse population demographics, healthcare settings, and resource availability [8, 9, 12]. Our investigation into the efficacy of the Onsite Syphilis Ab Combo Rapid test prototypes Fd and Ff reveals promising potential for these tools in enhancing syphilis screening and diagnosis, particularly in settings requiring immediate results. The high sensitivity and specificity exhibited by both prototypes Fd and Ff align with the indispensable need for reliable syphilis testing methods. A sensitivity of 100% is noteworthy, as it signifies the tests' capability to correctly identify individuals with syphilis, thereby facilitating timely intervention and reducing the risk of complications and further transmission [8, 9]. The specificity of the tests, standing at 98.8% for Fd and 99.6% for Ff, is equally crucial. High specificity minimizes the likelihood of false-positive results, which can lead to unnecessary stress and treatment in patients [10]. We observed notable discrepancies when compared with CMIA, RPR, TPPA, and the Determine™ Syphilis TP kit, particularly in the context of the reverse algorithm for syphilis screening. The rapid tests, despite their convenience and speed, demonstrated instances of strong positive results even in samples that were non-reactive according to CMIA. This discrepancy is crucial,

considering CMIA's high sensitivity and specificity, and it suggests that rapid tests might yield false positives. Therefore, in a reverse algorithm where CMIA is the initial screening tool, our findings emphasize the need for follow-up confirmatory testing to validate rapid test results.

Moreover, the role of CMIA as a primary screening method is reinforced due to its automated and more accurate nature, minimizing subjective errors common in manual testing. However, the occurrence of false positives in rapid tests, potentially due to cross-reactivity with other infectious or non-infectious diseases, highlights the complexity of syphilis diagnosis and the risk of misinterpretation [4, 8, 13–18]. This is especially in areas with prevalent conditions that could cause immunological cross-reactivity, raising the possibility of misdiagnosis [2, 4, 7, 9]. Furthermore, the study underscores the importance of comprehensive clinical assessment in interpreting rapid test results. This includes a detailed patient history and consideration of other potential non-infectious causes, such as autoimmune diseases, which could lead to false positives in syphilis rapid tests. Thus, clinical decision-making should not solely rely on rapid test outcomes but should consider the wider clinical context, including patient history, risk factors, and symptoms indicative of syphilis.

Moreover, it is important to consider the role of other infectious or non-infectious diseases that can cause false-positive results in syphilis rapid tests. Conditions such as HIV/AIDS [14], where immune system alterations can lead to atypical antibody responses; Lyme disease, caused by Borrelia burgdorferi [7]; Hepatitis B and C [2, 4, 19]; autoimmune disorders like systemic lupus erythematosus (SLE) and rheumatoid arthritis [12, 13, 15]; and physiological changes during pregnancy [6] can all influence test outcomes. Moreover, the phenomenon of biological false positives (BFPs) in serological tests, even when the infection is absent, adds another layer of complexity to syphilis diagnosis. These BFPs, resulting from various causes including acute infections and chronic conditions, complicate the interpretation of syphilis serological tests [12, 15].

Correlation analysis emphasizes the robust capacity of the Onsite Syphilis Ab Combo Rapid Test, particularly prototype Fd, to consistently display reactivity across a diverse range of RPR titer levels. The dominant strong positive (++) reactions observed across almost all titer levels highlight the reliability and precision of this testing prototype. Its potential as a trustworthy diagnostic tool is evident, offering clear and straightforward interpretation for clinicians in varied RPR titer scenarios. Prototype Fd stood out for its heightened sensitivity, especially in higher titer ranges, a capability that underlines its utility in various syphilis stages. The nuanced performance of prototype Ff at intermediate titers, however, suggests its applicability in settings where distinguishing between the stages of infection or monitoring treatment response is crucial. These insights underscore the importance of selecting a testing strategy aligned with the clinical scenario's specific needs, enhancing patient management and treatment outcomes.

Moreover, both prototypes Fd and Ff of the Onsite Syphilis Ab combo rapid test have demonstrated significant efficacy in detecting TPPA titers. Although both prototypes displayed remarkable consistency across most titer levels, prototype Fd exhibited a marginally higher sensitivity at the 160 and 1280 titer levels. However, the overall differences are minimal, and both prototypes can be deemed highly effective and reliable for syphilis detection based on the given TPPA titer levels. Equally compelling was the strong correlation of the rapid tests with reactive TPPA results. The tests' capacity to parallel the TPPA's sensitivity bolsters confidence in their role as dependable stand-ins for more conventional, labor-intensive methodologies. This is particularly salient in resource-constrained environments where access to standard confirmatory tests is limited, pointing to the broader implications of our findings for global health equity.

In a comprehensive evaluation by Herring et al. (2006), nine rapid syphilis tests were assessed across eight diverse laboratory sites, revealing promising performance characteristics.

Sensitivities and specificities ranged from 84.5% to 97.7% and 84.5% to 98%, respectively, relative to traditional reference standards like the *Treponema pallidum* haemagglutination (TPHA) or the TPPA tests [8]. These findings are particularly encouraging, indicating that these rapid tests, despite their operational simplicity, do not significantly compromise diagnostic accuracy. However, when these results are juxtaposed with the performance of the Onsite Syphilis Ab combo Rapid Test, certain nuances emerge. The Onsite Syphilis Ab Combo Rapid Test, especially the prototypes Fd and Ff, has demonstrated a consistent strong positive reaction across various RPR titer levels, an attribute that might be understated in the multi-centre evaluation study. This consistency is crucial, especially in high-prevalence settings, as it underscores the test's reliability in diverse epidemiological landscapes. Further assessments, such as those conducted by Diaz et al. (2004) and Bazzo et al. (2017), reinforce these observations, noting sensitivities up to 100% and specificities between 91.5% and 100% for various rapid tests [9, 10]. These high-performance metrics, particularly the tests' ability to maintain high specificity, are crucial to minimize false-positive results that could lead to unnecessary treatment and emotional distress [8].

Yet, the utility of these tests extends beyond their diagnostic accuracy. As Herring et al. (2006) highlight the operational characteristics of these tests, particularly their ease of use and the stability of results play a significant role in their applicability in real-world settings [8]. This ease of use was also echoed in the findings from Brazil, where professionals found the tests simple to administer and interpret [10]. Such operational advantages are particularly salient for the Onsite Syphilis Ab combo rapid test, which, due to its robust design, offers reliable performance even in non-laboratory environments, making it an invaluable asset in outreach settings and resource-constrained areas. However, a study into the Determine™ Syphilis TP kit effectiveness, particularly in outreach settings for female sex workers, indicated a sensitivity shortfall [18]. While the test's ease of use and rapidity were confirmed, its reduced sensitivity, especially in populations with a high prevalence of previously treated cases, suggests that reliance on this single rapid test might be insufficient. This underscores the importance of considering the specific epidemiological and demographic context when selecting a rapid syphilis testing strategy.

The robust performance of these rapid tests could be transformative in certain contexts. For instance, in prenatal care settings, the ability to swiftly and accurately diagnose syphilis in pregnant women means immediate treatment can be administered, significantly reducing the risk of congenital syphilis and associated complications [6]. Moreover, in resource-limited settings, where access to traditional laboratory facilities and skilled personnel is restricted, these tests offer a practical solution, requiring minimal training and infrastructure. However, the findings also bring to light the challenges associated with rapid syphilis tests. The regulatory classification of these tests as high-risk underscores the importance of stringent quality control and assurance measures. Ensuring that these rapid tests undergo rigorous evaluation before they are widely implemented is essential to maintain trust in their performance and to safeguard public health.

Furthermore, while the tests' ease of use and quick turnaround time are advantageous, it is vital to consider the implications of potential false positives due to cross-reactivity with other conditions. Continuous monitoring and supplementary testing strategies may be necessary to confirm diagnoses and prevent inappropriate treatment.

This study, while providing valuable insights into the efficacy of rapid diagnostic tests for syphilis, acknowledges several limitations. Firstly, the diversity and size of the sample group, although inclusive of syphilis patients, healthy individuals, pregnant women, and people with conditions such as HIV, HBV, and HCV, may not fully represent the broader population affected by syphilis, potentially limiting the generalizability of the findings. Secondly, the focus

on cross-reactivity was confined to specific conditions, omitting other infectious or non-infectious diseases that could impact test outcomes, possibly affecting the assessment of test specificity and sensitivity. The study's reliance on the CMIA method as the sole standard for comparison may not offer a comprehensive evaluation of the rapid tests' performance across different diagnostics. Furthermore, the cross-sectional nature of the study precludes the ability to observe the progression of syphilis over time or to evaluate test performance across various stages of the disease. The necessity for manual interpretation of the rapid tests introduces an element of subjectivity, highlighting the need for objective, automated reading solutions to ensure consistent result interpretation. Acknowledging these limitations is vital for understanding the context and scope of the study's contributions to improving syphilis diagnosis and management.

## Conclusion

The Onsite Syphilis Ab Combo Rapid test (prototypes Fd and Ff) have demonstrated potential as valuable assets in syphilis screening programs. Their high sensitivity, specificity, and predictive values indicate their reliability in accurately diagnosing syphilis, crucial for effective disease management and control. While their introduction into healthcare settings could be revolutionary, it necessitates a cautious approach, emphasizing quality assurance and the consideration of local epidemiology and resources. Further studies exploring their effectiveness in diverse populations and settings, as well as long-term outcomes, would be instrumental in solidifying their role in global health initiatives.

## Supporting information

**S1 Fig. Discrepancy results determined by the Onsite Syphilis Ab Combo Rapid Test (prototypes Fd and Ff) and the Determine™ Syphilis TP kit.**
(DOCX)

## Acknowledgments

We are grateful to thank the Thammasat University Hospital (TUH) for supplying blood samples.

## Author Contributions

**Conceptualization:** Sirinart Chomean, Chollanot Kaset.

**Data curation:** Palakorn Puttaruk, Phakawat Khamsophar, Waraphorn Fukpo.

**Formal analysis:** Chollanot Kaset.

**Investigation:** Sirinart Chomean, Palakorn Puttaruk, Phakawat Khamsophar, Waraphorn Fukpo, Chollanot Kaset.

**Methodology:** Chollanot Kaset.

**Visualization:** Chollanot Kaset.

**Writing – original draft:** Chollanot Kaset.

**Writing – review & editing:** Sirinart Chomean.

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
