## [Decision Letter · Decision Letter 0]

1 Apr 2024

PONE-D-23-43206Evaluation of rapid diagnostic test kits for detection of Treponema pallidum antibodyPLOS ONE

 Dear Dr. Kaset,

Thank you for submitting your manuscript to PLOS ONE. After careful consideration, we feel that it has merit but does not fully meet PLOS ONE’s publication criteria as it currently stands. Therefore, we invite you to submit a revised version of the manuscript that addresses the points raised during the review process.

We look forward to receiving your revised manuscript.

Kind regards,

Anoop Kumar, Ph.D.

Academic Editor

PLOS ONE

Journal Requirements:

2. Please note that PLOS ONE has specific guidelines on code sharing for submissions in which author-generated code underpins the findings in the manuscript. In these cases, all author-generated code must be made available without restrictions upon publication of the work. 

Please review our guidelines at https://journals.plos.org/plosone/s/materials-and-software-sharing#loc-sharing-code and ensure that your code is shared in a way that follows best practice and facilitates reproducibility and reuse.

"This study was supported by the Thammasat University Research Unit in Medical Technology and Precision Medicine Innovation. We are grateful to thank the Thammasat University Hospital (TUH) for supplying blood samples."

7. Please include a copy of Table 1 which you refer to in your text on page 9.

Reviewers' comments:

Reviewer's Responses to Questions

**Comments to the Author**

1. Is the manuscript technically sound, and do the data support the conclusions?

Reviewer #1: Yes

Reviewer #2: Partly

2. Has the statistical analysis been performed appropriately and rigorously? 

Reviewer #1: Yes

Reviewer #2: Yes

3. Have the authors made all data underlying the findings in their manuscript fully available?

Reviewer #1: Yes

Reviewer #2: Yes

4. Is the manuscript presented in an intelligible fashion and written in standard English?

Reviewer #1: Yes

Reviewer #2: Yes

5. Review Comments to the Author

Reviewer #1: The manuscript entitled “Evaluation of rapid diagnostic test kits for detection of Treponema pallidum antibody” is well written, structured and presents a comprehensive evaluation of two prototypes of the Onsite Syphilis Ab Combo Rapid Test (Fd and Ff), comparing their performance with the established chemiluminescent microparticle immunoassay (CMIA) method. The research addresses a significant gap in the field by assessing the efficacy and reliability of rapid syphilis testing, particularly in settings with limited resources where prompt and accurate diagnostics are essential. The reported sensitivity, specificity, positive predictive values (PPV), negative predictive values (NPV), and accuracies of the rapid test prototypes are well-documented and demonstrate their effectiveness as diagnostic tools. One notable strength of the study is the inclusion of diverse patient samples, including syphilis patients, healthy individuals, and cases with potential cross-reactions. This approach enhances the generalizability of the findings and highlights the utility of the rapid tests in real-world clinical scenarios.

Overall, this article provides valuable insights into the performance of the Onsite Syphilis Ab Combo Rapid Test prototypes and their potential integration into current diagnostic algorithms.

Reviewer #2: Dear Authors,

Thank you for your study aiming to assess the accuracy and reliability of two new versions of the The Onsite Syphilis Ab Combo Rapid Test that demonstrated high sensitivity and specificity comparable to CMIA methods.

Please, consider the suggestions below to improve on this good study.

Materials and Methods

Page 5, line 108.

“Population Study”.

I think Study population or Study participants would be a much-preferred Subheading here.

Page 6, line 113 -115

“A total of 450 serum samples were collected from Thammasat University Hospital (TUH). These were classified into three groups: 200 reactive samples for T. pallidum antibodies obtained from syphilis patients…”

Please, how and with what were these samples classified? How was the diagnosis of ‘syphilis’ made in your study participants before samples were collected from them?

Please, give more details on your data collection procedures and sample collection and processing.

Methods

Page 6, line 124-125

“Each test utilized 40 µL of serum followed by 40 µL of the provided buffer.” If after 15-minute incubation at room temperature, results were visually interpreted, it then shows that that the buffer contains the recombinant antigen of the TP bacterium. If this is the case, please indicate that the buffer contains the antigen, or that both buffer and the recombinant antigen were added also.

Correlation analysis of The Onsite Syphilis Ab combo Rapid Test with CMIA, RPR and TPPA test

Page 8, line 170-171

“Onsite Syphilis Ab Combo Rapid Test were grouped into W+, 2+, and 4+”. Please explain what W+ stands for and rationale behind the grouping into +W, 2+, and 4+.

Results

Page 9, line 196

“…as illustrated in Table 1”.

Please, where is the Table 1 mentioned here?

Also, consider preparing a three-column table displaying overall sensitivity, specificity, predictive values, and likelihood ratios of The Onsite Syphilis Ab combo Rapid Test using CMIA as the standard.

Measures of Diagnostic accuracy (Or Diagnostic Parameters) Estimated values (%) 95% Confidence intervals

Sensitivity

Specificity

PPV

NPV

PLR

NLR

Accuracy

Page 12, line 244

“…illustrated that its performs effectively across various…”

Please, correct to “…illustrated that it performs effectively across various…”

Page 15, line 272-276

“In conclusion, the Onsite Syphilis Ab combo Rapid Test (prototype Ff) demonstrates a

high degree of efficacy in detecting syphilis infections…”

I do not understand why your conclusion tend to project the performance of Prototype Ff over Fd while in your analysis, Prototype Fd out-performed Prototype Ff. I would suggest you reverse this conclusion in favour of Prototype Fd. Prototype Fd consistently gave strong positive reactions (100%) when compared with Ff (87.5%) against different TPPA tires.

Please, what do you think are the major limitations of this study? Please, add this before your conclusion.

Reference

Page 24, line 465, reference no 15

“…Wilson C, Peterman TA. 2019. Frequency and characteristics of biological false-positive test results…”

Please remove 2019 if it’s not part of the journal article title.

6. PLOS authors have the option to publish the peer review history of their article (what does this mean?). If published, this will include your full peer review and any attached files.

Reviewer #1: No

Reviewer #2: **Yes: **Dr. Anthony A. Iwuafor

---

## [Author Response · Author response to Decision Letter 0]

5 Apr 2024

Response to Reviewers

Dear Editor and Reviewers,

We are writing in response to the comments and suggestions provided on our manuscript entitled "Evaluation of Rapid Diagnostic Test Kits for Detection of Treponema pallidum Antibody," submitted to Plos One with manuscript number PONE-D-23-43206. We deeply appreciate the time and effort the reviewers have invested in evaluating our work, and we are grateful for the insightful feedback that has significantly contributed to the enhancement of our manuscript.

We have carefully reviewed and considered all the comments and have made corresponding revisions to our manuscript. Below is a summary of the major changes and our responses to the specific comments:

Materials and Methods

Page 5, line 108.

“Population Study”. 

I think Study population or Study participants would be a much-preferred Subheading here.

Response to Reviewer: Thank you for the suggestion. We have revised the text accordingly, as per Page 5, line 108.

Page 6, line 113 -115

“A total of 450 serum samples were collected from Thammasat University Hospital (TUH). These were classified into three groups: 200 reactive samples for T. pallidum antibodies obtained from syphilis patients…”

Please, how and with what were these samples classified? How was the diagnosis of ‘syphilis’ made in your study participants before samples were collected from them? 

Please, give more details on your data collection procedures and sample collection and processing.

Response to Reviewer: Syphilis diagnosis is based on the patient’s history, physical examination, laboratory testing, and sometimes radiology. However, many people with syphilis do not have any symptoms or have only minor symptoms and do not realize that anything is wrong. Identifying asymptomatic infection, especially among pregnant women, through screening using laboratory tests and treatment of positive cases will prevent further transmission and adverse pregnancy outcomes and congenital syphilis.

The available laboratory tests for syphilis include direct detection methods (i.e., dark-field microscopy, direct fluorescent antibody test, and nucleic acid amplification test), serology (treponemal and non-treponemal tests), and examination of cerebrospinal fluids. Regarding our study, the 450 serum samples collected from Thammasat University Hospital were classified into three groups based on reactivity for T. pallidum antibodies. Specifically, 200 samples were obtained from the positive group using the CMIA method, following standard clinical diagnostic procedures that included clinical assessment and serological tests such as the rapid plasma reagin (RPR) test and Treponema pallidum particle agglutination assay (TPPA). We have revised the text accordingly, as per Page 6, line 118-121.

Methods

Page 6, line 124-125

“Each test utilized 40 µL of serum followed by 40 µL of the provided buffer.” If after 15-minute incubation at room temperature, results were visually interpreted, it then shows that that the buffer contains the recombinant antigen of the TP bacterium. If this is the case, please indicate that the buffer contains the antigen, or that both buffer and the recombinant antigen were added also.

Response to reviewer: Thank you for your insightful comment regarding the composition of the buffer used in our study with the Onsite Syphilis Ab Combo Rapid Test, prototypes Fd and Ff. We appreciate the opportunity to clarify this aspect of our methodology.

We would like to clarify that the buffer provided with the rapid test kits does not contain the recombinant antigen of the Treponema pallidum (Tp) bacterium. Instead, the buffer's role is to facilitate the migration of the sample across the test strip by capillary action. The buffer composition typically includes 1% Bovine Serum Albumin (BSA), 0.5% sucrose, 0.05% Tween 20, and is diluted in Phosphate-Buffered Saline (PBS). These components serve as a running buffer, ensuring optimal conditions for the detection reaction without interfering with the specificity or sensitivity of the test.

The principle of the test, as detailed in our manuscript (PAGE 6, LINE 124-135 AND PAGE 7, LINE 136 -143), is based on the interaction between the antibodies present in the serum sample and the recombinant Tp antigens conjugated with colloidal gold, which are already pre-applied on the colored conjugate pad of the test cassette. This interaction occurs independently of the buffer composition, with the antigen-antibody complex then migrating towards the test and control lines to provide a visual result.

We hope this explanation adequately addresses your concern and clarifies the role of the buffer in the rapid test kits used in our study. We have amended our manuscript to ensure this explanation is explicitly stated, to prevent any misunderstanding regarding the test components and their respective functions.

Thank you once again for your valuable feedback, which has helped enhance the clarity and accuracy of our study's methodology.

Correlation analysis of The Onsite Syphilis Ab combo Rapid Test with CMIA, RPR and TPPA test

Page 8, line 170-171 “Onsite Syphilis Ab Combo Rapid Test were grouped into W+, 2+, and 4+”. Please explain what W+ stands for and rationale behind the grouping into +W, 2+, and 4+.

Response to reviewer: Thank you for your inquiry regarding the grading system used for the Onsite Syphilis Ab Combo Rapid Test.

In our study, the grading of positive results from the Onsite Syphilis Ab Combo Rapid Test was categorized into W+, 2+, and 4+. This grading system represents the strength of the positive band observed during the test, compared to the control band. Specifically:

W+ indicates a weak positive result

2+ indicates a moderate positive result

4+ indicates the strongest positive result

The purpose of this grading system is to provide a qualitative assessment of the intensity of the reaction observed on the test band. By categorizing results into different grades, we can compare the performance of the test with other testing methods. This helps us determine whether the two testing kits, which interpret reactions visually, exhibit concordance in reaction strength.

Results

Page 9, line 196

“…as illustrated in Table 1”. 

Please, where is the Table 1 mentioned here?

Also, consider preparing a three-column table displaying overall sensitivity, specificity, predictive values, and likelihood ratios of The Onsite Syphilis Ab combo Rapid Test using CMIA as the standard.

Measures of Diagnostic accuracy (Or Diagnostic Parameters) Estimated values (%) 95% Confidence intervals

Sensitivity 

Specificity 

PPV 

NPV 

PLR 

NLR 

Accuracy 

Response to Reviewer:

We apologize for the oversight. Table 1 was inadvertently omitted from the main manuscript. We have now inserted Table 1 into the manuscript as per your suggestion, and it can be found on Page 10, Lines 230-232. Thank you for bringing this to our attention, and we appreciate your thorough review.

Table 1 Evaluation Summary of Rapid Diagnostic Test Kits for Treponema pallidum Antibody Detection

Diagnosis Performance The Onsite Syphilis Ab combo rapid test 

 Fd Ff

Sensitivity % (95% Cl) 100.00 (98.17-100.00) 100 (98.17-100.00)

Specificity % (95% Cl) 98.80 (96.53-99.75) 99.60 (97.79-99.99)

PPV % (95% Cl) 1.64 (0.54-4.88) 4.76 (0.70-26.13)

NPV % (95% Cl) 100.00 (98.52-100.00) 100 (98.53-100.00)

PLR (95% Cl) 83.33 (27.06-256.63) 250.00 (35.35-1767.68)

NLR (95% Cl) 0 0

Accuracy % (95% Cl) 98.00 (97.30-99.59) 99.60 (98.47-99.96)

Page 12, line 244

“…illustrated that its performs effectively across various…”

Please, correct to “…illustrated that it performs effectively across various…”

Response to Reviewer:

Thank you for your keen observation. We have promptly corrected the error in the manuscript following your comment on Page 13, Line 275. The phrase now accurately reads, "...illustrated that it performs effectively across various..." We appreciate your attention to detail and apologize for any confusion caused by the oversight.

Page 15, line 272-276

“In conclusion, the Onsite Syphilis Ab combo Rapid Test (prototype Ff) demonstrates a 

high degree of efficacy in detecting syphilis infections…”

I do not understand why your conclusion tend to project the performance of Prototype Ff over Fd while in your analysis, Prototype Fd out-performed Prototype Ff. I would suggest you reverse this conclusion in favour of Prototype Fd. Prototype Fd consistently gave strong positive reactions (100%) when compared with Ff (87.5%) against different TPPA tires. 

Response to Reviewer:

We sincerely apologize for any confusion caused by the previous conclusion. We appreciate your insightful feedback and have revised the conclusion accordingly to better reflect the comparative performance of Prototype Fd and Prototype Ff.

New Conclusion:

Prototype Fd showed a marginally higher consistency in eliciting strong positive reactions across all TPPA titer levels, suggesting its slightly superior sensitivity in detecting syphilis antibodies. Prototype Ff, while also highly effective, presented minor variability at certain titer levels, which could allow for nuanced interpretation in specific clinical scenarios. Both prototypes prove valuable for syphilis screening, with Fd offering a slight advantage in uniform sensitivity, and Ff providing flexibility in response interpretation. This comparative analysis underscores the importance of selecting an appropriate diagnostic tool based on the clinical context and the need for confirmatory testing to ensure diagnostic accuracy. As Page 16, Line 340-311

We believe this revised conclusion better captures the nuanced performance of both prototypes and provides a balanced assessment of their strengths and limitations. Thank you for guiding us towards this improvement.

Please, what do you think are the major limitations of this study? Please, add this before your conclusion.

Response to Reviewer:

In addressing the major limitations highlighted in the reviewer's comment, we've added a comprehensive section before the conclusion to discuss these aspects. This addition aims to transparently acknowledge the constraints of our study, enhancing its contextual understanding.

Firstly, we acknowledge the limitation regarding the diversity and size of our sample group. While our sample included a range of participants, including syphilis patients, healthy individuals, and those with conditions potentially leading to cross-reactivity, it may not fully capture the broader demographic affected by syphilis. This limitation might impact the generalizability of our findings to all populations.

Secondly, our investigation into cross-reactivity was specifically focused on HIV, HBV, and HCV, without encompassing other infectious or non-infectious diseases that might influence the rapid tests' outcomes. This selective approach could potentially limit the comprehensiveness of our assessment regarding test specificity and sensitivity.

Moreover, our reliance on the CMIA method as the sole standard for comparison might not provide a fully rounded evaluation of the rapid tests’ performance across diverse diagnostic contexts. This singular comparison basis may restrict our understanding of the tests' efficacy in different settings.

Additionally, the cross-sectional nature of our study limits our ability to observe syphilis progression over time or assess test performance across the disease's various stages. This aspect could be crucial for understanding the dynamics of syphilis infection and the tests' diagnostic utility throughout the disease course.

Lastly, the manual interpretation of the rapid tests introduces a subjective element to our study. This subjectivity underscores the necessity for objective, automated reading solutions to ensure consistent and reliable result interpretation.

We hope this response addresses the concerns raised and provides a clearer picture of the steps taken to acknowledge and discuss the study's limitations within the manuscript.

Revised conclusion has been added as indicated on Page 21, Lines 430-435 and Page 22, 436-444 of our manuscript.

Reference

Page 24, line 465, reference no 15

“…Wilson C, Peterman TA. 2019. Frequency and characteristics of biological false-positive test results…” Please remove 2019 if it’s not part of the journal article title.

Response to Reviewer:

Thank you for bringing this to our attention. We have removed the year "2019" from Reference No. 15 in accordance with your suggestion. We appreciate your careful review of our manuscript.

---

## [Decision Letter · Decision Letter 1]

25 Apr 2024

Evaluation of rapid diagnostic test kits for detection of Treponema pallidum antibody

PONE-D-23-43206R1

Dear Dr. Kaset, 

We’re pleased to inform you that your manuscript has been judged scientifically suitable for publication and will be formally accepted for publication once it meets all outstanding technical requirements.

Kind regards,

Anoop Kumar, Ph.D.

Academic Editor

PLOS ONE

Additional Editor Comments (optional):

Reviewers' comments:

Reviewer's Responses to Questions

**Comments to the Author**

1. If the authors have adequately addressed your comments raised in a previous round of review and you feel that this manuscript is now acceptable for publication, you may indicate that here to bypass the “Comments to the Author” section, enter your conflict of interest statement in the “Confidential to Editor” section, and submit your "Accept" recommendation.

Reviewer #2: All comments have been addressed

2. Is the manuscript technically sound, and do the data support the conclusions?

Reviewer #2: (No Response)

3. Has the statistical analysis been performed appropriately and rigorously? 

Reviewer #2: (No Response)

4. Have the authors made all data underlying the findings in their manuscript fully available?

Reviewer #2: (No Response)

5. Is the manuscript presented in an intelligible fashion and written in standard English?

Reviewer #2: Yes

6. Review Comments to the Author

Reviewer #2: (No Response)

7. PLOS authors have the option to publish the peer review history of their article (what does this mean?). If published, this will include your full peer review and any attached files.

Reviewer #2: **Yes: **Dr. Anthony A. Iwuafor

---

## [Editor Report · Acceptance letter]

15 May 2024

PONE-D-23-43206R1 

PLOS ONE

Dear Dr. Kaset, 

I'm pleased to inform you that your manuscript has been deemed suitable for publication in PLOS ONE. Congratulations! Your manuscript is now being handed over to our production team.

Kind regards, 

on behalf of

Dr. Anoop Kumar 

Academic Editor

PLOS ONE